# Can weak segmentation labels from natural images aid digital pathology-based cancer classification?

**Eu Wern Teh** ETEH@UOGUELPH.CA and **Graham W. Taylor** GWTAYLOR@UOGUELPH.CA

*University of Guelph, Vector Institute for Artificial Intelligence*

*Vector Institute for Artificial Intelligence*

## Abstract

We explore the theme of utilizing weak segmentation labels from natural images to aid models perform better in Digital Pathology. A critical challenge with the medical image domain is the high cost of obtaining image labels from medical experts. On the other hand, it is considerably cheaper in time and money to obtain image labels from a layperson in the natural image domain. We show that general segmentation labels obtained from non-experts on a natural image dataset can boost performance on two cancer classification datasets (CRC and PCam). These results suggest that segmentation labels from natural images may not only benefit cancer classification but other future DP tasks.

**Keywords:** Transfer Learning, Digital Pathology

## 1. Introduction

Digital Pathology (DP) is a field that involves the analysis of microscopic images. Supervised learning has made progress in DP for cancer classification and segmentation tasks in recent years; however, it requires a large amount of labels to be effective (Srinidhi et al., 2020). Unfortunately, labels from medical experts are scarce and extremely costly. On the other hand, it is relatively cheap to obtain weak labels[1] from a layperson in the natural image domain. We aim to transfer knowledge from natural images to DP where labels are scarce.

Transfer learning via pre-training allows a model to learn features from one task and transfer them to another task to improve generalization. Models are commonly pre-trained with classification labels from ImageNet (Deng et al., 2009), and these pre-trained models have helped to boost the performance of many computer vision tasks (Huh et al., 2016). However, transfer learning via semantic segmentation labels is an unexplored area, especially from natural images to DP. We hypothesize that models pre-trained on these weak segmentation labels can help DP models achieve higher performance when compared to randomly initialized models.

## 2. Experiments

We evaluate the downstream tasks with various amounts of labels to simulate human annotation scarcity by domain experts on two datasets: CRC (Kather et al., 2016) and PCam (Veeling et al., 2018). CRC is a colorectal cancer dataset that consists of 625 images per class, and PCam is a breast cancer dataset that consists of 131,072 images per class. We train our downstream models using the same settings as Teh and Taylor (2020) and report our results as mean accuracy and standard deviation over ten runs with random initializations.

---

1. We define weak labels as labels obtained from the natural image domain

| Initialization/Method | CRC dataset | | | PCam dataset | | |
|---|---|---|---|---|---|---|
| | $N_c$ | $R\%$ | Accuracy (%) | $N_c$ | $R\%$ | Accuracy (%) |
| Random | 12 | 2 | $61.62 \pm 3.79$ | 1,000 | 0.76 | $79.37 \pm 1.33$ |
| ImageNet | | | $\mathbf{79.16 \pm 2.83}$ | | | $\mathbf{83.91 \pm 0.84}$ |
| Classification | | | $62.98 \pm 3.70$ | | | $76.61 \pm 1.83$ |
| Segmentation | | | $67.34 \pm 3.80$ | | | $82.11 \pm 1.28$ |
| Random | 25 | 4 | $64.12 \pm 3.73$ | 2,000 | 1.53 | $81.26 \pm 1.47$ |
| ImageNet | | | $\mathbf{83.42 \pm 1.57}$ | | | $\mathbf{86.12 \pm 0.60}$ |
| Classification | | | $69.16 \pm 3.87$ | | | $80.95 \pm 1.18$ |
| Segmentation | | | $73.34 \pm 2.17$ | | | $86.11 \pm 1.13$ |
| Random | 50 | 9 | $77.34 \pm 2.11$ | 3,000 | 2.29 | $84.09 \pm 1.24$ |
| ImageNet | | | $\mathbf{88.06 \pm 1.73}$ | | | $86.52 \pm 0.63$ |
| Classification | | | $78.00 \pm 1.25$ | | | $82.50 \pm 1.10$ |
| Segmentation | | | $81.40 \pm 2.05$ | | | $\mathbf{87.80 \pm 0.67}$ |

Table 1: Accuracy of our model trained with $R\%$ of data in four different pre-trained settings on the CRC dataset and PCam dataset. $N_c$ denotes the number of examples per class. Models for classification and segmentation are pre-trained on the Pascal VOC 2012 dataset (Everingham et al., 2012).

### 2.1. Pre-training on Pascal VOC 2012 dataset

We pre-train a multi-label classification model and a semantic segmentation model on the Pascal VOC 2012 dataset (Everingham et al., 2012). There are a total of 10,582 training images and 1,449 validation images in Pascal VOC 2012. We use a randomly initialized ResNet-34 model to train the classification model and a randomly initialized DeepLabV2 model with ResNet-34 backbone for segmentation pre-training (He et al., 2016; Chen et al., 2017). We use the 2D cross-entropy loss for segmentation training and the binary cross-entropy loss for multi-label classification training. We optimize both models for 40,000 iterations using Stochastic Gradient Descent with a learning rate of 2.5e-4, a momentum of 0.9, and a weight decay of 5e-4. We also use a polynomial decay rate of $\lambda(1 - \frac{\text{iter}}{\text{max\_iter}})^{0.9}$, where $\lambda$ is the base learning rate. We augment the dataset via random cropping ($321 \times 321$), random horizontal flipping (0.5), and random re-scaling (0.5 to 1.5). After pre-training, we initialize the downstream models with the corresponding ResNet-34's weights.

### 2.2. Discussion

Models pre-trained on Pascal segmentation labels outperform both the randomly initialized models and models pre-trained on Pascal classification labels (Table 1). However, models pre-trained on Pascal classification labels hardly improve over the random baseline. The Pascal VOC 2012 dataset (10k images) is relatively small compared to the ImageNet dataset (1 Million images). On the PCam dataset, the performance of segmentation pre-trained models is comparable to the ImageNet pre-trained models even though they are pre-trained on a smaller number of images. However, the ImageNet pre-trained models outperform the segmentation pre-trained models in all three subsets on the CRC dataset. We speculate that the performance differences may be related to a low number of training images in the pre-

training phase. We hypothesize that the segmentation pre-trained models can outperform ImageNet pre-trained models given the same number of pre-trained images.

## 3. Conclusion

We show that models pre-trained with weak segmentation labels provided by non-experts can outperform randomly initialized models in cancer classification. We anticipate that DP researchers will consider using cheap segmentation labels from natural images as an additional source of supervision to improve the performance of DP models. Larger-scale generic segmentation datasets will support future work in this direction.

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
