# OpenReview forum: "Can weak segmentation labels from natural images aid digital pathology-based cancer classification?"
_MIDL.io/2021/Conference/Short — Submitted to MIDL 2021_

### Official Review · Reviewer_2FTy · 2021-05-01

**Confidence:** 5
**Final Rating:** 2

**Summary:**

The authors aim to show that classification networks for digital pathology can be initialized on segmentation tasks from natural image datasets. They train both ResNet-34 classification models and segmentation models with a ResNet-34 backbone (DeepLabV2) on the Pascal VOC 2012 dataset. These pre-trained ResNet-34 weights are then used for transfer learning to a new digital pathology classification task.

**Strengths:**

The idea seems interesting and the motivation for utilizing natural image datasets for pre-training models for transfer learning to medical images has shown to be a successful strategy in many prior works. The utilization of segmentation datasets has not been explored much (as stated by the authors) as most prior works utilized pre-trained models on ImageNet, i.e. a classification dataset.
The paper is also well written and clearly presented.

**Weaknesses:**

While the motivation of the authors is clear, the experimental results do not support the idea. The ImageNet pre-trained models outperform the models pre-trained on a segmentation task on all the targeted digital pathology classification tasks.
The same hyperparameters are utilized to pre-train both the classification and segmentation networks. Likely different hyperparameters are needed for the models to converge well on each of the tasks.

**Deanonymize Review:**

no

**Justification Of The Rating:**

While the idea is interesting, the experimental results are not supporting the motivation of this work. The authors hypothesize that models pre-trained on larger segmentation datasets would outperform ImageNet pre-trained methods. I suggest the authors back up that hypothesis with experiments before pursuing a conference publication.

**Paper Type:**

methodological development

**Special Issue:**

no

---

### Meta-Review · Area_Chair_Asum · 2021-05-11

**Recommendation:** Reject
**Confidence:** 4

**Metareview:**

I agree with the reviewer that while not really novel the particular application of the idea is of interest. Unfortunately, the results do not support the hypothesis that segmentation pre-training on Pascal might be more useful for biomedical images than default ImageNet class pre-training. Interestingly the authors argue that this could be due to the differences size of the datasets, but fail to run this crucial experiment. A more systematic choice of hyper parameter and more experiments are necessary before this work can    be published. Unfortunately, I therefore have to recommend a rejection.

---

### Decision · Program_Chairs · 2021-05-11

Reject